# Interferon-Stimulated Gene 15 Knockout in Mice Impairs IFNα-Mediated Antiviral Activity

**DOI:** 10.3390/v14091862

**Published:** 2022-08-24

**Authors:** Chen Li, Wen-Feng He, Long-Xi Li, Jing Chen, Guo-Qing Yang, Hong-Tao Chang, Hui-Min Liu

**Affiliations:** 1College of Life Science, Henan Agricultural University, Zhengzhou 450002, China; 2College of Animal Veterinary Medicine, Henan Agricultural University, Zhengzhou 450002, China

**Keywords:** pseudorabies virus, ISG15 knockout mice, in vivo, susceptibility, type I IFN, impair

## Abstract

Type I interferon (IFN) plays an important role in the host defense against viral infection by inducing expression of interferon-stimulated genes (ISGs). In a previous study, we found that porcine interferon-stimulated gene 15 (ISG15) exhibited antiviral activity against PRV in vitro. To further investigate the antiviral function of ISG15 in vivo, we utilized ISG15 knockout (ISG15^-/-^) mice in this study. Here, we demonstrate that ISG15^-/-^ mice were highly susceptible to PRV infection in vivo, as evidenced by a considerably reduced survival rate, enhanced viral replication and severe pathological lesions. However, we observed no significant difference between female and male infected WT and ISG15^-/-^ mice. Moreover, ISG15^-/-^ mice displayed attenuated antiviral protection as a result of considerably reduced expression of IFNβ and relevant ISGs during PRV replication. Furthermore, excessive production of proinflammatory cytokines may be closely related to encephalitis and pneumonia. In further studies, we found that the enhanced sensitivity to PRV infection in ISG15^-/-^ mice might be caused by reduced phosphorylation of STAT1 and STAT2, thereby inhibiting type I IFN-mediated antiviral activity. Based on these findings, we conclude that ISG15 is essential for host type I IFN-mediated antiviral response.

## 1. Introduction

Pseudorabies virus (PRV), also called Aujeszky’s disease virus or suid herpesvirus 1 (SuHV-1), is a large, enveloped DNA virus that belongs to the subfamily Alphaherpesvirinae [1]. Although pigs are the natural reservoir for the virus, a wide variety of mammals are also susceptible to PRV [2]. Recent studies revealed that humans might also be a potential host of PRV [3,4,5]. PRV infection poses a major threat to the pig industry, and various types of vaccines are usually used to control the disease [6]. Although vaccines can eliminate viral infection, emerging PRV variants have caused frequent outbreaks of PRV infection since late 2011 [7]. Thus, novel antiviral agents need to be developed as a to complement vaccination.

Type I interferon (IFN) represents the first line of defense to combat viral infections. Interferon-stimulated gene 15 (ISG15), an IFN-α/β-inducible, ubiquitin-like molecule, has been reported to play an antiviral role in viral infection [8,9,10,11]. We previously reported that porcine ISG15 plays an antiviral role during PRV infection [1]. However, there are large differences among species with respect to the role of ISG15 in host–virus interactions; unlike human ISG15, which stabilizes USP18, the murine orthologue does not exert such activity [12]. In vitro studies in mouse cells have demonstrated an antiviral role of ISG15 during several viral infections [13,14,15,16], although there are some reports of viruses displaying no enhanced replication when ISG15 is deficient [16,17]. Knocking down ISG15 in human cells has also suggested an antiviral role of ISG15 during infection with numerous viruses [13,18,19,20], whereas other studies have suggested no role at all [21,22]. Furthermore, mice lacking ISG15 exhibit enhanced susceptibility to some but not all viruses [23,24], whereas ISG15 deficiency in humans has been reported to enhance viral resistance [12,25,26,27]. According to these results, ISG15 deficiency may actually increase resistance to severe viral infections.

Type I IFN stimulation has been shown to inhibit PRV replication due to the induction of several ISGs that function as general antivirals. Among these, we found that ISG15 significantly upregulated and further demonstrated an antiviral role of ISG15 during PRV infection in vitro [28], although nothing is known about the exact role of ISG15 in type I IFN-mediated antiviral response against PRV in vivo. Here, for the first time, we examine the antiviral effect of ISG15 in vivo and show that mice lacking ISG15 were highly susceptible to PRV infection. Moreover, ISG15 deficiency in mice resulted in attenuated IFNα-mediated antiviral protection by considerably reducing the expression of IFNβ and relevant ISGs. We found that excessive production of proinflammatory cytokines in ISG15^-/-^ mice may be closely related to encephalitis and pneumonia. Finally, we found that the enhanced sensitivity of ISG15^-/-^ mice to PRV infection might be associated with reduced phosphorylation of STAT1 and STAT2.

## 2. Materials and Methods

### 2.1. Virus and Reagents

Pseudorabies virus (PRV) strain QXX was propagated in porcine kidney 15 (PK15) cells (CCL-33, ATCC) and titered by plaque assay. The primary antibodies used for Western blotting included rabbit anti-STAT1, anti-STAT2 (Proteintech, Wuhan, China), rabbit anti-phospho-Tyr701 STAT1, rabbit anti-phospho-Tyr690 STAT2 (Abcam, Cambridge, UK), anti-ISG15 and anti-β-actin (Proteintech).

### 2.2. Mouse Experiments

C57BL/6N (WT) and ISG15^-/-^ mice were purchased from Cyagen Biosciences, Inc. (Guangzhou, China). The mice were mated, bred and genotyped in the animal facility. The genotype of bred mice was determined by PCR. Animal experiments were performed in accordance with protocols approved by the Use of National Research Center for Veterinary Medicine (Permit 20180521047).

Mice were represented in both the control and infected groups. Mice (both male and female) aged 7–9 weeks were intraperitoneally inoculated with 50–200 plaque-forming units (PFU) of PRV diluted in 40 μL sterile DMEM. Mice were monitored daily for body weight and clinical symptoms. Brain, lung and spleen tissues were collected to determine viral loads, mRNA level of ISGs and expression of proinflammatory cytokines for histological analysis.

### 2.3. PCR, Quantitative Real-Time PCR and Western Blotting

Total RNA from the tissues was extracted using RNAiso Plus reagent (Takara, Dalian, China) and was reverse transcribed to cDNA using a PrimeScript^TM^ RT reagent kit with gDNA Eraser (Takara) according to the manufacturer’s protocols. The genes used in this study were amplified by PCR using Prime STAR^®^ Max DNA Polymerase (Takara) or with real-time PCR using TB Green Premix Ex Taq II (Takara). Quantitative real-time PCR (RT-qPCR) was carried out utilizing SYBR Green PCR Master Mix (Promega, Beijing, China). The mRNA level of each target gene was normalized to that of β-actin mRNA.

Protein was extracted from the homogenized tissues using a lysis buffer, and total protein concentration was determined with a BCA protein assay kit (Beyotime Biotechnology, Shanghai, China). An equal amount (30 μg) of protein from tissues of WT and ISG15^-/-^ mice was resolved on 10% SDS-PAGE gels and transferred to a PVDF membrane (Pall Corporation, Ann Arbor, MI, USA). The membrane was probed with corresponding antibodies, followed by horseradish peroxidase-conjugated anti-rabbit IgG (Servicebio, Wuhan, China). The protein bands were visualized by chemiluminescence using ECL (Beyotime Biotechnology, Shanghai, China). The intensities of the protein bands were quantified using Image J software and standardized against β-actin. All data are presented as three independent experiments in duplicate.

### 2.4. Enzyme-Linked Immunosorbent Assay (ELISA)

Protein concentration of the cytokines, including interleukin (IL)-6, IL-1β and tumor necrosis factor (TNF)-α, from organs of infected mice were measured by ELISA using mouse IL-6, IL-1β and TNF-α kits (Meimian, Nanjing, China). The absorbance was read with an automated ELISA plate reader at 450 nm.

### 2.5. Histopathological Analysis

On day 4 post infection, mice were sacrificed, and lungs and brains were collected for histological analysis. Tissues were fixed and dehydrated and embedded in paraffin, followed by sectioning (4–5 μm) and staining with hematoxylin–eosin (HE). Pathological changes were determined through observation of the morphologic characteristics.

### 2.6. Statistical Analysis

Statistical analyses were performed using GraphPad Prism. The difference between groups was calculated using Student’s t-tests, and the differences were considered to be significant when * *p* < 0.05, ** *p* < 0.01 or *** *p* < 0.001. The standard errors of the mean (SEM) of at least three independent experiments are shown for each data.

## 3. Results

### 3.1. Mice Lacking ISG15 Are More Susceptible to PRV Infection

PRV can infect a wide variety of mammals, including rodents; therefore a mouse model has been widely used to study PRV pathogenesis [29]. To determine whether ISG15 also exhibits antiviral activity against PRV in vivo, ISG15-deficient mice were generated on a C57BL/6N background by targeting ISG15 for deletion using CRISPR-Cas9 technology. Seven-week-old female and male mice were injected intraperitoneally with PRV, and their clinical symptoms, body weight and survival rate were monitored daily. The results show that the ISG15^-/-^ mice displayed typical neurological symptoms, including considerably reduced activity and pruritus at 3 days post infection (dpi), with death beginning at 4 dpi, whereas WT mice developed only mild symptoms at 5 dpi under the same conditions (Figure 1A). On the other hand, the ISG15^-/-^ mice lost weight more rapidly than WT mice by day 6, and the surviving WT mice continued to lose weight (Figure 1B).

Next, the mortality of wild-type (WT) and ISG15^-/-^ mice was monitored for 10 days after PRV infection. We found that all ISG15^-/-^ mice died within 6 days after PRV infection, whereas 53.3% of the WT mice remained alive under the same conditions. As illustrated in Figure 1C, ISG15^-/-^ mice had a survival rate of 0, whereas WT mice had a survival rate of 53.3%, suggesting that ISG15^-/-^ mice are more sensitive to PRV than WT mice. Furthermore, PRV loads in the brain, lungs and spleen were assayed at 4 dpi. The viral loads of the ISG15^-/-^ mice were markedly higher than those in the same tissues of WT mice (Figure 1D). Moreover, no significant difference between female and male mice was observed in terms of body weight, survival rate or viral load (data not shown). The above results indicate that ISG15 deficiency enhances susceptibility to PRV infection in vivo.

### 3.2. ISG15 Deficiency Promotes PRV Infection Pathogenicity in Mice

PRV infection mainly leads to neurological and respiratory symptoms, and encephalitis is a contributing factor to animal death [30]. To detect the degree of pneumonia and encephalitis in infected mice, we histopathologically analyzed the brains and lungs at 4 dpi. As shown in Figure 2A, the brains of ISG15^-/-^ mice showed a substantial number of necrotic neurons and more necrotic Purkinje cells than those in WT mice (Figure 2A). Furthermore, microgliosis and hyperemia were more obvious in the brains of ISG15^-/-^ mice. Histological analyses of infected lungs showed increased inflammatory cell infiltration, severe congestion and higher levels of lung tissue impairment in ISG15^-/-^ mice in comparison with WT mice (Figure 2B). These results indicate that ISG15 deletion aggravated virus-induced pathogenicity during PRV infection in vivo.

### 3.3. ISG15 Deficiency Impairs Type I IFN Production

Type I IFN (IFN-α/β) signaling is critical for host restriction of viral infection [31]. To test whether ISG15 is involved in type I IFN-mediated antiviral innate immune response in vivo, we first compared the body weight between WT and ISG15^-/-^ mice after infection with PRV with or without IFNα treatment. Starting on day 6, the infected mice pretreated with IFNα started to recover their body weight, albeit at a significantly lower rate compared to WT mice (Figure 3A). Moreover, ISG15^-/-^ mice pretreated with IFNα exhibited a dramatically reduced mortality rate, with no significant difference compared with WT counterparts (Figure 3B). We also compared the body weight and survival rate between female and male mice, with no significant difference observed (data not shown).

To analyze the tissue distribution of the PRV genome in organs of infected mice, we detected DNA of PRV-gE in different tissues after PRV infection for 4 dpi. Higher levels of PRV-gE DNA were found in infected tissues from ISG15^-/-^ mice in comparison with those from WT mice (Figure 3C). Additionally, expression of IFNβ was significantly down-regulated in various tissues from ISG15^-/-^ mice in response to PRV infection (Figure 3C). To further test the effect of ISG15 on IFNβ production induced by PRV infection, we first examined the association between ISG15 expression patterns and PRV replication in vivo. We found that the high expression of ISG15 in brains of infected WT mice inhibited PRV growth, whereas ISG15 deficiency significantly promoted PRV replication (Figure 3D). Moreover, the PRV load in IFNα-treated ISG15^-/-^ mice showed a much higher fold than that in WT mice, suggesting that ISG15 plays an important role in promoting IFN-mediated antiviral response. Thus, we next detected the levels of IFNβ in brains, lungs and spleens between WT and ISG15^-/-^ mice infected with PRV untreated or treated with IFNα. We found that the expression of IFNβ was considerably reduced in infected ISG15^-/-^ mice pretreated with IFNα as compared to that in WT counterparts (Figure 3E). These findings were further confirmed by RT-qPCR (Figure 3F).

Taken together, these data suggest that ISG15 deficiency impairs type I IFN production and promotes PRV replication.

### 3.4. ISG15 Deficiency Suppresses the Expression of Some ISGs in Response to PRV Infection

The data reported above show that ISG15 knockout considerably reduced IFNβ production. Therefore, we analyzed the expression of several key ISGs, including IFIT1, OAS1 and Mx1 in various tissues of infected mice. Results indicate that the ISG15 knockout resulted in impaired expression of IFIT1, OAS1 and Mx1 in the brain during PRV infection (Figure 4A), which was further confirmed by RT-qPCR (Figure 4B). To verify this finding, the expression of these ISGs in the lungs and spleens were also detected, and similar results were obtained by RT-PCR and RT-qPCR (Figure 4C–F). These findings suggest that ISG15 may be involved in antiviral immune responses by modulating type I IFN signaling.

### 3.5. Mice Lacking ISG15 Produce Excessive Inflammatory in Response to PRV Infection

Cytokines are crucial in combating viral infection and are involved in the regulation of immune and inflammatory responses, including interleukin 1β (IL-1β), interleukin-6 (IL-6) and tumor necrosis factor alpha (TNF-α) [32]. The mRNA levels of IL-1β, IL-6 and TNF-α were markedly elevated in brains, lungs and spleens of infected ISG15^-/-^ mice relative to WT mice (Figure 5A–E). After treatment with IFNα, the mRNA levels of these cytokines were significantly reduced, whereas their expression levels were still higher in ISG15^-/-^ mice than WT mice (Figure 5A–E). Furthermore, the protein concentrations of IL-1β, IL-6 and TNF-α were measured using ELISA assay, and a similar tendency was observed in protein levels. As shown in Figure 5G, IL-1β, IL-6 and TNF-α protein levels were significantly higher in the serum of ISG15^-/-^ mice than those in WT mice with or without IFNα treatment (Figure 5G). These data suggest that ISG15^-/-^ mice displayed a more severe inflammatory response than WT mice, indicating that ISG15 deficiency leads to excessive production of inflammatory cytokines.

### 3.6. ISG15 Deficiency Potentiates Viral Replication by Blocking STAT1/STAT2 Phosphorylation

Because ISG15 deficiency impaired IFNβ expression in response to PRV infection, as shown in Figure 3, we speculated that ISG15 may affect IFNβ production by targeting phosphorylated STAT1 (pSTAT1) or/and STAT2 (pSTAT2). Therefore, we tested whether ISG15 deficiency affects the expression of pSTAT1 and pSTAT2 in tissues of infected mice by Western blot. We found that the expression of pSTAT1 and pSTAT2 was significantly reduced in brains of ISG15^-/-^ mice in comparison with WT counterparts. Moreover, IFNα stimulation promoted the expression of pSTAT1 and pSTAT2 in the brains of infected mice (Figure 6A,B). Similar results were observed in the lungs of infected mice (Figure 6C,D). These data reveal that the ISG15 deficiency inhibits the expression of pSTAT1 and pSTAT2, which is most likely responsible for reduced IFNβ production during PRV infection, suggesting that ISG15 deficiency impairs host antiviral activity against PRV by attenuating the expression of pSTAT1 and pSTAT2.

## 4. Discussion

The function of ISG15 in herpesvirus remains an area of active investigation [28,33]. In vitro studies in PK15 cells revealed that ISG15 plays an antiviral role in the context of PRV infection. To examine whether ISG15 exhibit an antiviral response against PRV in vivo, we used ISG15^-/-^ mice and found that mice lacking ISG15 were more sensitive to PRV infection relative to WT mice (Figure 1). ISG15^-/-^ mice showed a decreased body weight, increased PRV growth and increased disease severity (Figure 1 and Figure 2). Our findings are in line with the previous results showing that mice lacking ISG15 exhibit enhanced susceptibility to challenge with many viruses [17,23,24]. Furthermore, PRV infection caused more severe damage in the brain and lung tissues of ISG15^-/-^ mice than those of WT mice, indicating that ISG15 deficiency could worsen histopathological changes.

It has been reported that PRV infection could activate the immune response in the rodent brain, including type I IFN and inflammatory cytokines, to allow PRV to establish a persistent infection [34]. Although considerable progress had been made in terms of understanding the pathogenesis of PRV [35], the function of ISG15 in type I IFN-mediated antiviral response during PRV infection is not fully understood. Here, we found that mice lacking ISG15 had significantly downregulated expression of IFNβ and several critical ISGs (Figure 3 and Figure 4), which may be responsible for the increased susceptibility of ISG15^-/-^ mice to PRV infection. Although inflammatory response is the first line of defense to prevent the spread of viral infections, uncontrolled inflammatory response usually leads to severe inflammation, which may cause damage to the host [29]. In this study, excessive production of cytokines, including IL6, IL-1β and TNF-α, was observed in infected ISG15^-/-^ mice, indicating that ISG15 is critically involved in antiviral response. Cytokines are the key regulators of host defense against viral infection; however, excessive inflammatory response induced by viral infection is associated with viral pathogenesis. It has been reported that a cytokine storm (excessive production of IL-6, IL-1β and TNFα) caused by highly pathogenic influenza virus infection can injure host organs, resulting in severe disease and even death [36,37]. Thus, it is very likely that the considerably elevated levels of inflammatory cytokines in infected ISG15^-/-^ mice contribute to the pathogenesis of PRV.

It has been reported that IFNα and IFNβ serve as homeostatic agents during inflammatory response [37,38]. Consistent with this view, we found that ISG15 knockout not only resisted IFNβ production but also increased expression of inflammatory cytokines. However, after IFNα treatment, the expression of cytokines was reduced in infected ISG15^-/-^ mice. These observations confirm that IFNα and IFNβ play a critical role in the control of inflammation. In addition, downregulated phosphorylation of STAT1 and STAT2 were observed in ISG15^-/-^ mice after infection with PRV in vivo. Therefore, we speculate that reduced pSTAT1 and pSTAT2 may be associated with elevated inflammatory response. Our observation indicates that mice lacking ISG15 blocked type I IFN signaling by inhibiting phosphorylation of STAT1 and STAT2, thereby preventing an antiviral response. We observed that the expression of pSTAT1 and pSTAT2 was reduced in the brains and lungs of infected ISG15^-/-^ mice, although similar reductions were not detected in other tissues. One possible interpretation for these observations is that an excessive inflammatory response in brains and lungs may be responsible for encephalitis and pneumonia caused by PRV infection. Further studies are needed to address this possibility.

## 5. Conclusions

Altogether, our findings indicate that ISG15 precisely regulates the host antiviral response by controlling STAT1/2 activation and IFNβ production. These results provide a new perspective on the functions of ISG15; moreover, the results of our study may provide a useful strategy for the prevention and control of this viral disease.

## Figures and Tables

**Figure 1 viruses-14-01862-f001:**
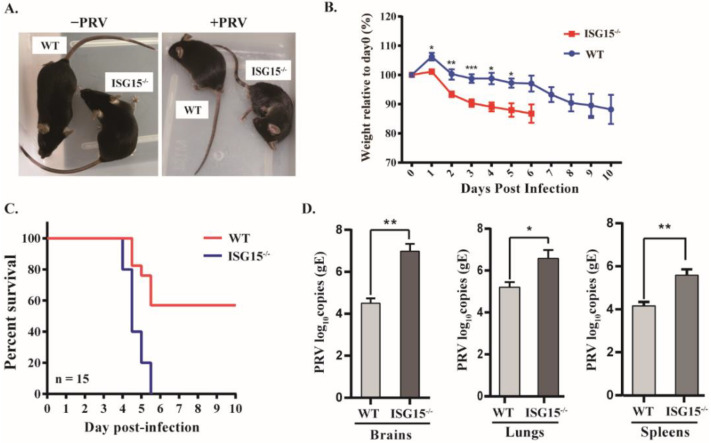
Mice lacking ISG15 are more susceptible to PRV than WT mice. (**A**) The clinical aspects of WT and ISG15-deficient (ISG15^-/-^) mice infected intraperitoneally with PRV are shown. (**B**,**C**) Body weight (**B**) and survival (**C**) were monitored for 10 dpi (WT: *n* = 15; ISG15^-/-^: *n* = 15). Data are shown as percent change of body weight relative to the starting weight on day 0 (mean ± SEM). Survival curves were compared using a log-rank test. (**D**) At 4 dpi, the brain, lung and spleen tissues of the infected mice were harvested and homogenized, and PRV gE gene copies were assayed by quantitative real-time PCR (RT-qPCR). PRV-infected ISG15^-/-^ mice vs. PRV-infected WT mice. * *p* < 0.05; ** *p* < 0.01; *** *p* < 0.001.

**Figure 2 viruses-14-01862-f002:**
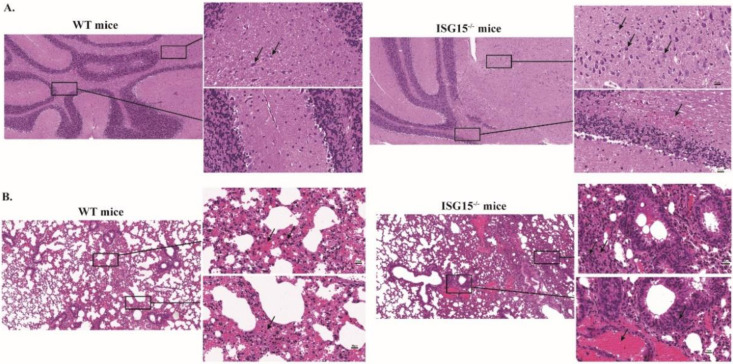
Histopathological examination of brains and lungs of infected ISG15^-/-^ and WT mice. WT (*n* = 10) and ISG15^-/-^ (*n* = 10) mice were intraperitoneally infected with PRV. At 4 dpi, the brains and lungs of infected mice were collected, sectioned and stained with hematoxylin-eosin. Magnified regions are indicated with rectangles in (**A**,**B**). Scale bar: 100 μm and 20 μm.

**Figure 3 viruses-14-01862-f003:**
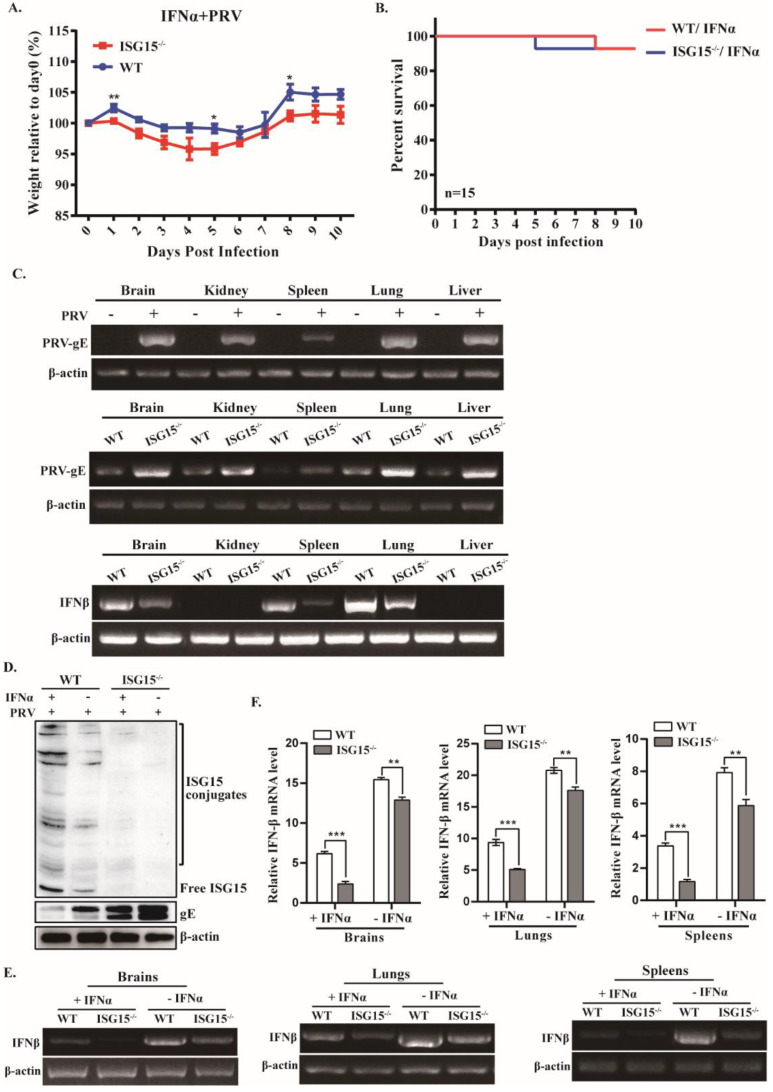
Expression of IFNβ is reduced in ISG15^-/-^ mice after PRV infection. (**A**,**B**) Body weight and survival rate of infected WT and ISG15^-/-^ mice with IFNα treatment were monitored for 10 days. (**C**) WT and ISG15^-/-^ mice were infected intraperitoneally with PRV for 4 days, followed by PCR for detection of viral DNA levels in the indicated tissues from infected mice and RT-PCR for detection of IFNβ levels in indicated organs of WT and ISG15^-/-^ mice (**C**). (**D**) The protein expression of ISG15 and PRV-gE in brains of WT and ISG15^-/-^ mice were analyzed by Western blotting. (**E**,**F**) IFNβ levels in the indicated tissues of infected mice were examined by RT-PCR (**E**) and RT-qPCR (**F**). The average results from three independent experiments are plotted. * *p* < 0.05; ** *p* < 0.01; *** *p* < 0.001.

**Figure 4 viruses-14-01862-f004:**
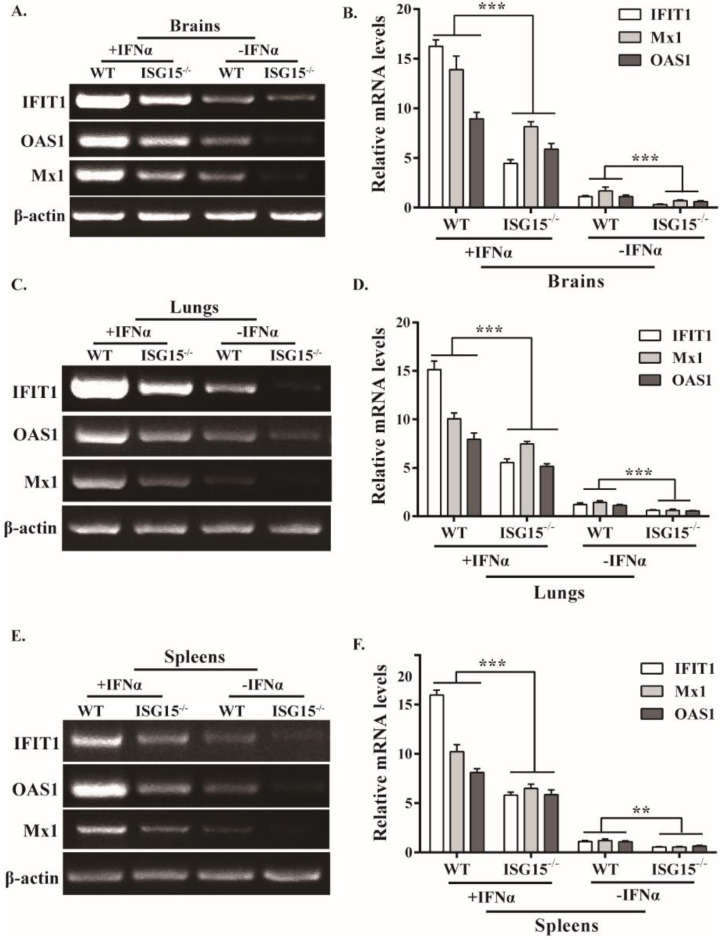
Several ISGs mRNA levels are downregulated in infected ISG15^-/-^ mice. WT and ISG15^-/-^ mice were intraperitoneally infected with PRV and pretreated or untreated with IFNα. At 4 dpi, the mRNA levels of IFIT1, OAS1 and Mx1 in brains (**A**), lungs (**C**) and spleens (**E**) of infected mice were examined by RT-PCR and RT-qPCR (**B**,**D**,**F**), respectively. ** *p* < 0.01; *** *p* < 0.001.

**Figure 5 viruses-14-01862-f005:**
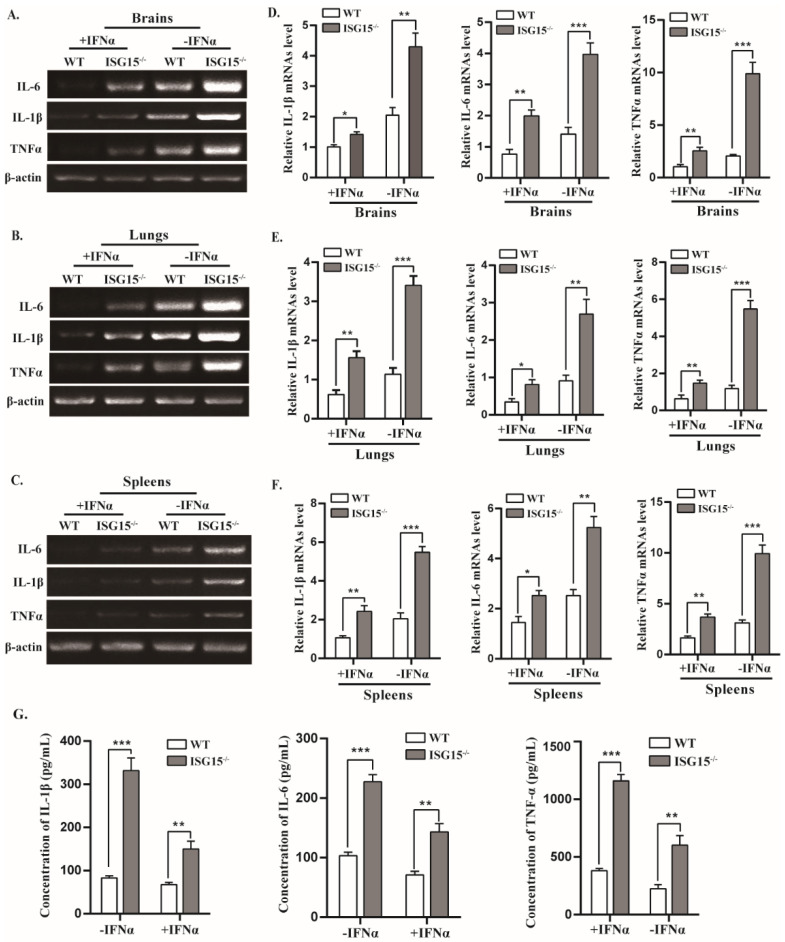
Expression of inflammatory cytokines was significantly increased in infected ISG15^-/-^ mice. (**A**–**F**) The RNA levels of IL-1β, IL-6 and TNFα in the indicated tissues from PRV-infected WT and ISG15^-/-^ mice by RT-PCR and RT-qPCR, respectively. The data were normalized to β-actin. (**G**) The protein levels were measured by ELISA. * *p* < 0.05; ** *p* < 0.01; *** *p* < 0.001.

**Figure 6 viruses-14-01862-f006:**
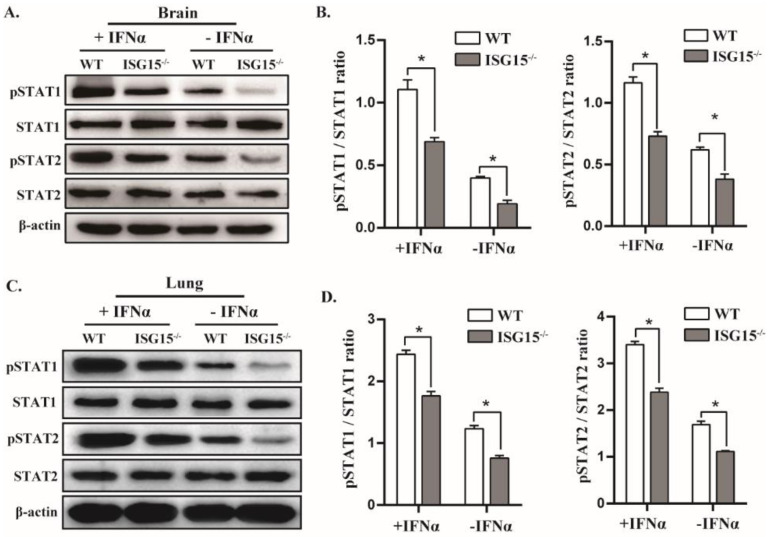
STAT1 and STAT2 phosphorylation are considerably reduced in infected ISG15^-/-^ mice. The tissues of infected WT and ISG15^-/-^ mice either pretreated or left untreated were harvested and analyzed by Western blotting with the indicated antibodies (**A**,**C**). Phosphorylated STAT1 or STAT2 levels relative to STAT1 or STAT2 were quantitated by densitometry and normalized to β-actin (**B**,**D**). The average of three independent replicates were plotted (means ± SD). * *p* < 0.05.

## Data Availability

All available data are presented in the article.

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
