# Peer review of "Interferon-Stimulated Gene 15 Knockout in Mice Impairs IFNα-Mediated Antiviral Activity"

_viruses, 2022, doi:10.3390/v14091862_

Round 1

Reviewer 1 Report

It is fine as is.

Author Response

We are very appreciated to your support.

Reviewer 2 Report

In this paper the authors aim to show that ISG15 is important for IFNα-mediated antiviral activity against Pseudorabies virus (PRV) in vivo by employing a ISG15 knockout (ISG15-/-) mouse model. The authors demonstrated that these ISG15-/- mice were more susceptible to PRV infection compared to WT mice unless they were pre-treated with IFNα. They also showed that the expression of ISGs in PRV infected ISG15-/- mice was severely abrogated which they found was due to a lack of STAT1/2 phosphorylation in the knockout mice. Finally, they determined that PRV infected ISG15-/- mice expressed more pro-inflammatory cytokines than the WT basally and with the addition of IFNα. Although the phenotype when investigated seems true, the rationale for how ISG15 impacts IFNα/b expression and thereby anti-viral responses was lacking. ISG15 is downstream of IFNa/b induction, so how ISG15 regulates IFNa/b expression? If the expression of IFNa/b is reduced in ISG15, consequently IFNAR-mediated signaling will be affected resulting in less STAT1/2 phosphorylation and less ISG expression. Thus, the effect of ISG15 on the STAT signaling pathway is indirect. But then how ISG15 controls the upstream, first arm of the type I IFN signaling pathway. It is not clear from the manuscript, and it was not discussed.

Comments

  1. Please include what the role of ISG15 is in pigs since they are the natural reservoir of PRV (around lines 41-42)
  2. Lines 57-58: “Moreover, ISG15 deficiency in mice displayed attenuated IFNa-mediated antiviral protection by greatly lowering the expression of IFNb and relevant ISGs.” I suggest rephrasing this sentence because it sounds strange to say that IFNa-mediated antiviral response was reduced due to less IFNb expression.
  3. Figure 1 A: This panel does not add to the take-away message of the figure. The figure legend says that this panel shows the clinical aspects of infected WT and mutant mice. Nothing can be seen on the photo besides the mice.
  4.  Figure 1C: The IFNα pre-treatment data should not be here. It is described later in the Results section. It should be moved and made into an additional panel for Figure 3.
  5.  Figure 1D: Is the Y-axis in log10 scale? 4 copies of viral genome per mg of tissue seem to be extremely low. ISG15 knockout results in only 2 viral DNA copy increase?
  6. Figure 2: please include stars, arrows or arrowheads to indicate where on your H&E stains you see the necrotic neurons and Purkinje cells, microgliosis and hyperemia. Results section 3.2 says that this is what the histological analysis shows in Fig 2A but it is not clear what can be seen on the photos.
  7.    Figure 3B: The comparison between viral load of WT and ISG15-/- in varying tissues is missing here, only uninfected or PRV infected is shown. This has to be addressed.
  8. Figure 3B, C, E: Using real-time qPCR would be better for quantitative comparison.
  9. Fig 3F and E: Why IFNa treatment causes less IFNb expression in WT mice (and in mutant)? Also, how does IFNa induce IFNb? IFNa and IFNb are induced by IRF3/IRF7, and then IFNa/b are secreted from the cells and induce IFNAR-mediated ISG expression.
  10.  Figure 5: please include male and female data for this figure to show there is no significant difference between sexes.

Comments about text/ suggested edits:

Abstract:

  • Line 15 “To further investigate the antiviral function of ISG15 in vivo, we generated mice with complete ISG15 deficiency.” According to the Methods section, the ISG15 mouse was purchased from a company, so the authors cannot say that they generated it.
  • Line 16: Change wording from “Here, we describe ISG15 Knockout mice were…” to “Here we describe (or demonstrate) that ISG15 knockout mice were…”
    • The sentence was awkwardly worded

Introduction:

  • Line 33: human should be changed to humans
  • Lines 36 – 37: this sentence is very hard to read, please consider changing the wording
  •  Lines 52 – 53: I suggest changing the wording of this sentence since it is hard to read.

Materials and Methods:

  • Line 66: Misspelling of titer. Change from tittered to titered
  • Line 84: add space between “andwas”

Results:
3.1:

·         Line 117:

o   “variety wide” should be changed to “wide variety”

o   Change from “thus, model” to thus, a model”

·         Line 118:

o   Change from “had” to “has”

3.2:

·         Line 151: Typo – should say “Figure 2A” not 2B

3.3:

·         Line 165: awkward wording – I suggest deleting “investigate the function of ISG15 in” to make the sentence less wordy.

·         Line 172: replace “gene” to “genome”.

·         Line 182: This is too bold a statement for where it is place in the paper. This should be stated in the conclusions.

·         Line 183: awkward wording, please change.

3.4:

·         Line 204: insert “was” between “Mx1” and “remarkedly”. Change “remarkedly” (misspelling) to “remarkably”

3.5:

·         Line 219: Change “obviously” to “markedly”

3.6:

·         Line 239: please indicate the figure and panel instead of stating “shown above”.

Discussion:

·         Line 276: misspelling – change “sever” to “severe”

Author Response

   Thank you for considering and reviewing our manuscript viruses-1844591 entitled “Interferon-stimulated Gene 15 Knockout Mice Impair IFNα-mediated antiviral activity”. We appreciate the reviewers’ comments and suggestions very much. Here we decide to submit our revised manuscript for the further reviewing. We have performed additional experiment suggested by the reviewer#2 and rephrased the manuscript; the revised manuscript has been prepared according to the formatting notes, and a revised clean copy with a revised marked copy is enclosed in this resubmission. Here we are addressing the reviewers’ queries and concerns in detail as follows
